# A Review of the Clinical Features and Management of Systemic Congenital Mastocytosis through the Presentation of An Unusual Prenatal-Onset Case

Valérie Larouche [1,*], Marie-Frédérique Paré [2], Pierre-Olivier Grenier [3], Anna Wieckowska [4], Eric Gagné [5], Rachel Laframboise [6], Nada Jabado [7] and Isabelle De Bie [8]

1  Department of Pediatric Hemato-oncology, Centre Hospitalier Universitaire de Quebec-Université Laval, Quebec, QC G1V4G2, Canada
2  Medecine Faculty, Laval University, Quebec, QC G1V4G2, Canada
3  Department of Dermatology, Centre Hospitalier Universitaire de Quebec-Université Laval, Quebec, QC G1V4G2, Canada
4  Departement of Pediatric, Centre Hospitalier Universitaire de Quebec-Université Laval, Quebec, QC G1V4G2, Canada
5  Department of Pathology, Centre Hospitalier Universitaire de Quebec-Université Laval, Quebec, QC G1V4G2, Canada
6  Department of Medical Genetics, Centre Hospitalier Universitaire de Quebec-Université Laval, Quebec, QC G1V4G2, Canada
7  Department of Pediatric Hemato-Oncology, Montreal Children's Hospital, McGill University Health Centre, Montreal, QC G1V4G2, Canada
8  Division of Medical Genetics, Department of Specialized Medicine, McGill University Health Centre, Montreal, QC G1V4G2, Canada
*  Correspondence: valerie.larouche.med@ssss.gouv.qc.ca

**Abstract:** Mastocytosis is a heterogeneous group of rare hematological disorders that can occur in infancy. We report a 16-year-old girl who presented with an aggressive form of systemic congenital mastocytosis, associated with a significant global developmental delay, deafness, and multiple anomalies. At 4 years of age, she developed a germinoma presenting as an invasive spinal mass. Extensive cytogenetic, metabolic, and molecular genetic studies that included whole-exome sequencing studies revealed a *KIT* alteration (NM_000222.3(KIT):c2447A > 7 pAsp816Val) and likely pathogenic variant in the DNA from peripheral blood and skin lesions. C-kit was also found to be overexpressed in the spinal tumor cells. We compared the features of this child to those of six previously reported pediatric patients with cutaneous mastocytosis, microcephaly, microtia, and/or hearing loss reported in OMIM as mastocytosis, conductive hearing loss, and microtia (MIM 248910), for which the etiology has not yet been determined. This report extends the currently recognized spectrum of KIT-related disorders and provides clues as to the potential etiology of a syndromic form of congenital mastocytosis. International efforts to understand the benefits of long-term targeted therapy with tyrosine kinase inhibitors for this KIT-altered rare disease should continue to be evaluated in clinical trials.

**Keywords:** congenital mastocytosis; KIT alteration; CNS tumor; hearing loss; microtia; tyrosine kinase inhibitor





## 1. Introduction

Mastocytosis (MIM 154800) is a rare hematological disorder characterized by abnormal mast cell proliferation. There is a spectrum of clinical manifestations, which varies by organ involvement. Cutaneous mastocytosis is the most frequent clinical presentation in children, with symptoms usually appearing within the first two years of life. Prognosis is variable depending on the presence of systemic involvement. Typically, children without systemic disease show improvement or resolution around the time of puberty, while adults present a high risk of chronic disease associated with systemic involvement or malignant

progression. Systemic congenital mastocytosis is extremely rare in children and carries a poorer prognosis [1–9]. According to the WHO classification, the diagnosis of systemic mastocytosis (SM) can be established if at least one major and one minor, or three minor criteria are fulfilled [8,10] (Table 1).

**Table 1.** Major and minor criteria used to establish a diagnosis of systemic mastocytosis [10].

| Major Criteria | Minor Criteria |
|---|---|
| Multifocal, dense infiltrates of MCs (≥15 mast cells in aggregates) detected in sections of bone marrow biopsies and/or sections of other extra-cutaneous organ(s). | 1. ≥25% of all mast cells are atypical cells on bone marrow smears or other extra cutaneous organs. |
| | 2. Detection of KIT point mutation at codon 816 or in other critical regions of KIT in bone marrow, or in another extra-cutaneous organ. |
| | 3. Mast cells in bone marrow, blood, or another extra-cutaneous organ express one or more of the following: CD2, and/or CD25, and/or CD30. |
| | 4. Serum total tryptase > 20 ng/mL (no other associated myeloid neoplasm). |

The vast majority of mastocytosis cases are caused by isolated somatic activating mutations in the proto-oncogene KIT, most frequently the p.D816V variant (reference sequence NP_000213.1) [11–13]. Acquired KIT receptor pathogenic variants are less frequent in children than in adults. Few familial cases of cutaneous mastocytosis have been reported [14–19], showing both autosomal recessive and dominant inheritance patterns with variable penetrance. Mastocytosis has also been described in association with familial gastrointestinal stromal tumors (GISTs; MIM 606764), which can be caused by pathogenic variants in the *SDHB*, *SDHC*, *PDGFRA*, and *KIT* genes [20–24]. The frequency of systemic involvement in mastocytosis affecting children and its outcome are still unknown [25].

Somatic KIT alterations can also be detected in germ cell tumors, specifically in testicular seminomas, ovarian dysgerminomas, and extragonadal germinomas, which all share the same etiology [26–30]. The development of subsequent cutaneous mastocytosis has been reported [31–35]. However, to our knowledge, the occurrence of pediatric mastocytosis followed by the development of a germ cell tumor has not yet been described, nor has the syndromic association of congenital mastocytosis and deafness as the possible underlying etiology of this clinical presentation.

Herein, we report a child who presented with severe systemic congenital mastocytosis associated with various anomalies, including deafness. This patient later developed a spinal germinoma. Extensive genetic investigations which included whole-exome sequencing revealed a KIT pathogenic variant (NM_000222.3(KIT):c2447A > 7 pAsp816Val) as a potential etiology.

## 2. Case Report

We report the case of a 16-year-old girl who is the first child of non-consanguineous French-Canadian parents. She has five healthy siblings. The pregnancy was uneventful; there were no maternal health complications or teratogenic exposure. She was born at term by spontaneous-assisted vaginal delivery. All birth parameters were at the 50th percentile for gestational age (birth weight 3980 g, height 50.5 cm, and head circumference 34.5 cm). APGAR scores were 7 [1] and 9 [5]. Neonatal examination revealed erythroderma associated with a tense cutaneous vesicle on the forehead, as well as a massive hepatosplenomegaly (Figure 1).

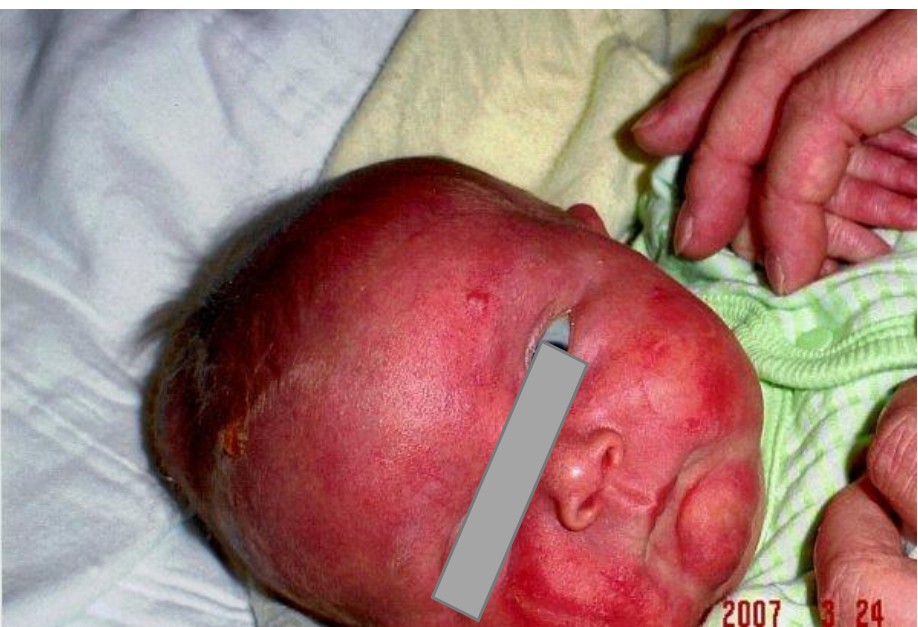

**Figure 1.** Erythroderma with a vesicle involving the forehead in the newborn period.

Other dysmorphic features included a large anterior fontanelle, triangular facies, a prominent forehead, midface hypoplasia, hypertelorism, small palpebral fissures, bilateral epicanthal folds, left esotropia, a short and upturned nose, a smooth philtrum, microstomia, and microtia. The rest of the physical examination was otherwise unremarkable.

Clinical investigations included the following: a normal complete blood count with smears; a negative septic workup including a TORCH screen; and liver function tests which showed a slight elevation of indirect bilirubin at 127 μmol/L (normal values rank between 1 and 16 μmol/L) and GGT at 133 UI/L (normal values rank between 4 and 18 UI/L). Other liver enzymes, albumin, and coagulation tests were normal. There was no hypoglycemia. Kidney function tests were normal. Beta-human chorionic gonadotropin, alpha-foetoprotein, alpha-1 antitrypsin, urinary vanillylmandelic (VMA) acid, and homovanillic acid (HVA) dosages were all within the normal range. Tryptase was elevated at 84.9 ug/L (normal values are usually between 3,8 and 11,4 ug/L), and subsequently increased to 122–147 ug/L. RT-PCR analysis for the FIP1L1-PDGFRA fusion product was negative.

An abdominal ultrasound did not reveal structural anomalies other than a large hepatosplenomegaly and a para-aortic soft-tissue mass measuring 2.2 × 1.5 cm. A gallium scan was negative for both neuroblastoma and bony lesions.

## 3. Pathological Evaluations

A skin biopsy demonstrated a massive dermal infiltration of mast cells. Bone marrow aspirate and a biopsy performed at three months of age showed interstitial mast cell infiltration with the expression of CD2 (minor criteria) consistent with mastocytosis. A concomitant liver biopsy confirmed hepatic mast cell infiltration complicated by periportal fibrosis and extramedullary haematopoiesis. Another bone marrow biopsy performed later at 2 years old confirmed the major criteria with the presence of multifocal, dense infiltrates of MCs (≥15 mast cells in aggregates) detected. Immunochemistry for CD117 (C-Kit) identified mast cells in the skin, liver, and bone marrow biopsies (Figure 2).

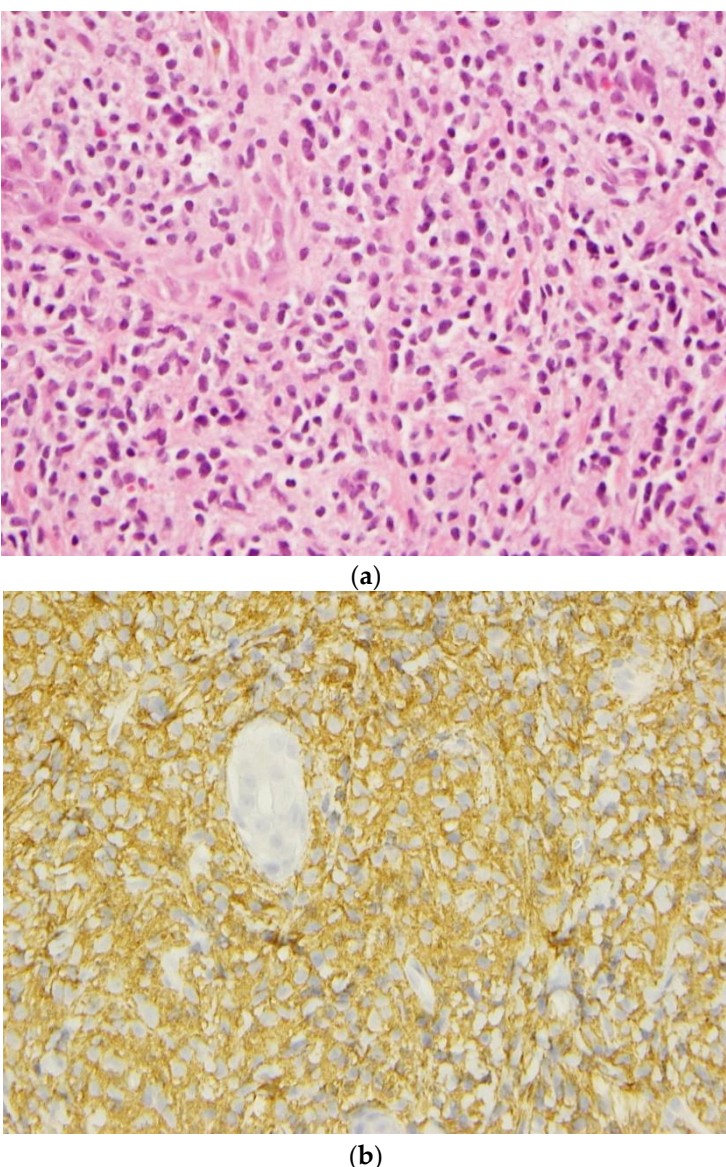

**Figure 2.** (**a**) Mast cell infiltration in the skin and (**b**) immunochemical stain for c-Kit/CD117 show a strong expression in the cytoplasmic membrane of the mastocytes.

## 4. Genetic Evaluations

Both a standard karyotype performed on bone marrow and a CGH microarray analysis (Agilent CGX$^{TM}$–HD $4 \times 180$ K oligo platform) of DNA extracted from leucocytes revealed a normal female chromosome complement. Specifically, bone marrow FISH staining did not reveal the presence of the Philadelphia chromosome (Nuc ish 9q34 (ABLx2) 22q11 (BCRx2). Plasma amino acids and urine organic acids were both normal as well as urine mucopolysaccharide and oligosaccharide screens. Whole-exome sequencing of leukocyte DNA only revealed a de novo KIT pathogenic variant (NM_000222.3(KIT):c2447A > 7 pAsp816Val), which was also present in fibroblasts cultured from a skin biopsy.

The newborn met the diagnostic criteria for disseminated mastocytosis with cutaneous, bone marrow, hepatic, and spleen involvement.

## 5. Clinical Evolution

By three months of age, signs of portal hypertension appeared on an abdominal Doppler ultrasound, revealing an aggressive systemic form of mastocytosis as per the classification criteria (C-criteria) of Valent et al. [36]. Malabsorption was also suspected

based on a mild decrease in vitamin K-dependent coagulation factors, a deceleration of weight gain, and diarrhea.

The patient started on multiple anti-histaminic medications, Lanzoprazole, multivitamins, and calcium supplementation. Prednisone was initiated at 3.5 months of age with some cutaneous response, but worsening hepatosplenomegaly, portal hypertension, and elevated tryptase levels persisted (Figure 3). High-dose prednisone led to a failure to thrive (weight and height below the 3rd percentile; head circumference remaining at the 50th percentile), osteoporosis, delayed bone age, adrenal insufficiency, and pseudotumor cerebri. Because of these complications, progressive weaning off prednisone was undertaken over several months. Cromoglycate sodium was initiated during prednisone weaning.

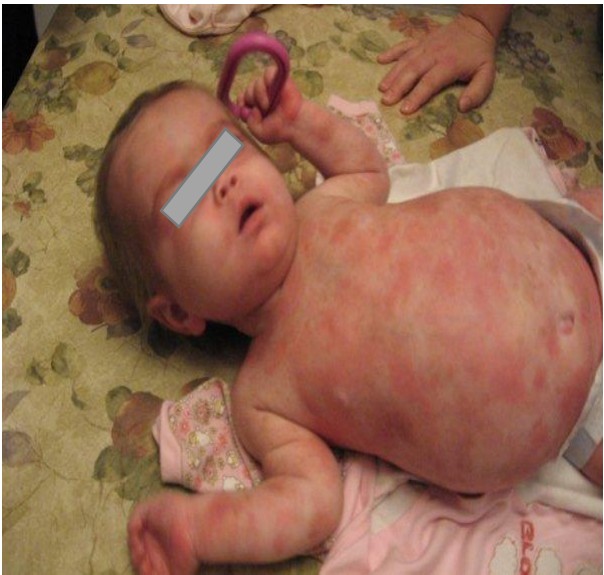

**Figure 3.** Erythematous plaques and patches, large abdomen from the massive hepatosplenomegaly, and strabismus are shown in this 10-month-old girl.

Imatinib administration was started at 18 months of age. Unfortunately, the treatment initiation coincided with cutaneous disease flare-ups likely caused by a massive mast cell degranulation, as well as a severe cutaneous toxicity reminiscent of Stevens-Johnson syndrome, and therefore needed to be discontinued (Figure 4).

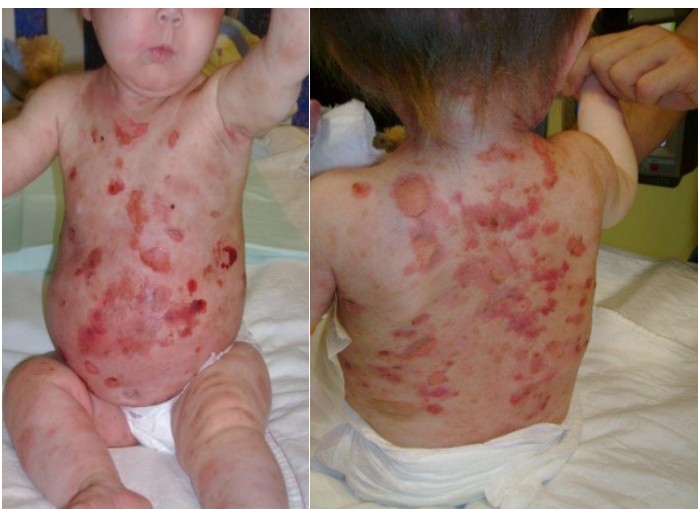

**Figure 4.** Bullae and vesicles, which appeared over the trunk when Imatinib was introduced at 18 months of age.

At 2 years of age, as part of the investigation into the failure to thrive, a gastroscopy did not reveal any inflammation or significant mast cell infiltration, and fecal elastase, fecal alpha-1-antitrypsin, and albumin were normal. A repeat liver biopsy showed massive mastocyte infiltration with advanced portal fibrosis and septa formation (stage F3/4), which was suggestive of a progression towards cirrhosis. A repeat bone marrow biopsy confirmed massive mastocyte infiltration. Treatment with interferon alpha at 600 000 units every 3 weeks was then started at 28 months of age for a period of 18 months. A liver biopsy performed 8 months after the initiation of interferon alpha confirmed a significant improvement in the portal fibrosis and a regression of the mastocyte infiltration. Unfortunately, treatment had to be discontinued due to major irritability and a suspicion of seizures (abnormal EEG despite the absence of clinical seizures).

At 4 years of age, the patient was admitted for increased irritability, apathy, and possible regression. The neurological examination showed a diminished right lower limb strength and clonus. A spine MRI revealed a voluminous, solid, and expansive enhancing intramedullary lesion of 5.2 cm at the conus, with concomitant medullary oedema (Figure 5). Alpha-foetoprotein and BHCG markers in the blood and the CSF were both negative.

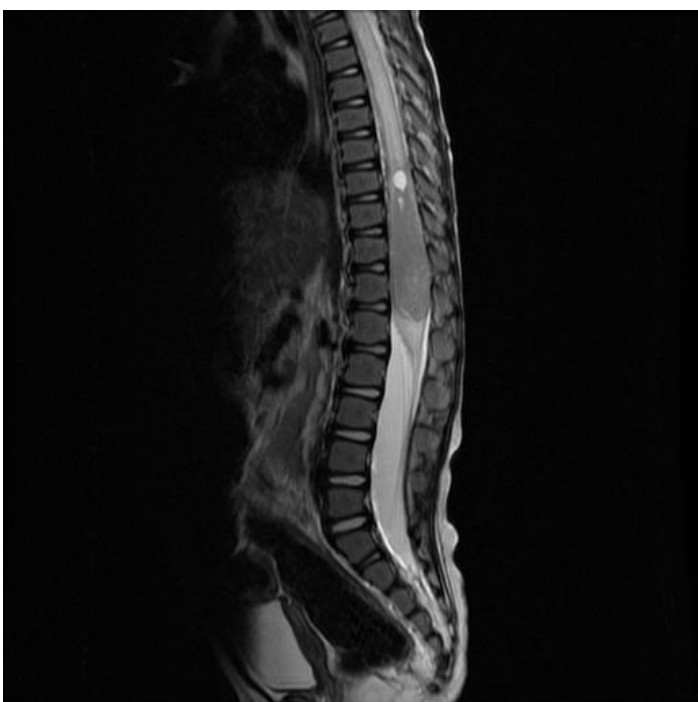

**Figure 5.** Large intramedullary tumor seen in the conus on MRI.

Because of intramedullary extension, only a near-total resection could be attained. A diagnosis of pure germinoma was confirmed by pathological analysis; the germinoma cells showed strong c-kit expression. A brain MRI was negative for parenchymal lesions but showed a diffuse thickening of the bone, which is compatible with mastocyte infiltration.

Unfortunately, the patient had a local relapse two months after the surgery, with the appearance of a 4.5 cm expansive spinal mass. She was treated with focal radiation therapy for a total of 12 Gy. To this day, subsequent imaging has revealed no recurrence so far.

Dasatinib treatment, a tyrosine kinase inhibitor, was initiated one month after the end of radiation therapy in 2011. Currently, the patient is on maintenance Dasatinib therapy in an attempt to prevent a tumor relapse or the development of additional tumors at other sites. Medication is currently well tolerated, despite initial gastrointestinal intolerance. Twelve years later, we have observed no failure of the treatment.

At around 5 years of age, she was also diagnosed with epilepsy and had convulsive episodes. She has been treated with Topiramate and to this day has had no other episodes.

On the neurological side, she also has significant hypotonia, cognitive and developmental delay, and a suspicion of autism. Despite clinical improvement in SM, the patient demonstrated severe global developmental delay. To this day, the patient is still unable to walk independently but can crawl. She is still non-verbal. Moderate bilateral hearing loss was confirmed from the age of 5 months with abnormal brainstem auditory-evoked potentials. However, the level of development of the child did not allow reliable conventional evaluations in soundproof booths. In addition, she does not tolerate hearing aids. Although eye contact remains poor to this day, ophthalmologic evaluation only revealed strabismus with left esotropia and amblyopia. A CT scan of the mastoids revealed normal external and internal auditory canals as well as inner-ear anatomy. The only significant findings were abnormal aspects of bone structures, compatible with mastocyte infiltration. A skeletal survey showed bilateral metaphyseal humeral sclerosis and delayed bone age; no other skeletal anomalies were reported. Orthodontic evaluation revealed generalised dental hypoplasia and multiple cavities.

Liver disease seemed to worsen over time. Liver ultrasounds showed persistent hepatosplenomegaly and signs of portal hypertension. A liver fibro scan performed at 9 years of age showed elevated liver stiffness. This finding could not be explained by liver mast cell infiltration alone, as other signs of SM improved. Real fibrosis and a progression to liver cirrhosis were thus suspected.

At 12 years of age, the patient presented with irritability. Liver enzymes and GGT increased from normal to values above 300 U/L persistently. Based on an elevated tryptase level of 195 ug/l, hepatic mastocytosis progression was suspected and the Dasatinib dose was increased. However, there was no improvement in liver enzymes. Further tests showed elevated IgG 24 g/L, increased antinuclear antibodies (1:2500), and markedly increased anti-liver/kidney microsomal antibodies (1:1024), suggestive of type 2 autoimmune hepatitis. Other etiologies were excluded. A liver biopsy was not performed due to a high bleeding risk in the patient. She responded to a cycle of corticosteroids followed by long-term azathioprine therapy, resulting in the normalisation of liver enzymes.

Repeated liver ultrasounds as well as a liver MRI performed at 13 years of age demonstrated signs of portal hypertension, signs of revascularisation, and a dysmorphic, nodular liver suggestive of real cirrhosis (and not only liver disease due to mast cell infiltration). As of now, she remains with a normal liver function, normal platelets, and has not developed esophageal varices.

## 6. Discussion

To our knowledge, this is the first report of a patient presenting with neonatal-onset aggressive SM, who then developed a spinal germinoma. The KIT pathogenic variant (NM_000222.3(KIT):c2447A > 7 pAsp816Val) is a likely etiology for both of these clinical manifestations. However, severe developmental delay and deafness are not classically associated with SM. A clinical entity in which mastocytosis and deafness are concurrently present has previously been described (MIM 248910). Six patients with cutaneous mastocytosis developing in childhood, as well as microcephaly, microtia, and/or hearing loss, have been reported so far [37–41] (Table 2). Three of these patients had congenital-onset cutaneous mastocytosis but none presented with systemic involvement. Our patient presented almost all of the criteria of this clinical entity, including hearing impairment and microtia. Growth retardation as well as seizures could theoretically be explained as medication side effects.

Neonatal SM is a rare presentation of congenital mastocytosis and carries a poor prognosis. Organ complications, malignant transformation, leukemia, anaphylaxis, and haemorrhage are some of the complications that can lead to death. Many fatalities related to bullous mastocytosis in early infancy were probably secondary to massive mast cell degranulation and shock [42].

**Table 2.** Clinical features in the literature and the present case.

| Clinical Manifestations | Wolach et al [38]. | Hennekam and Beemer [39]. | Salpetrio et al [37]. | Ina et al [41]. | Trevisan et al. [40]. | | Present Case |
|---|---|---|---|---|---|---|---|
| | | | | | Case 1 | Case 2 | |
| Cutaneous mastocytosis | + | + | + | + | + | + | + |
| Neonatal mastocytosis | + | + | − | + | − | − | + |
| Systemic mastocytosis | − | − | − | − | − | − | + |
| Hearing loss | + | + | − | + | + | + | + |
| Feeding problems | + | + | − | − | − | − | + |
| Hypotonia | + | + | + | − | − | − | + |
| Microcephaly | + | + | + | − | − | − | − |
| Mental retardation | − | + | + | − | − | − | + |
| Convulsion | − | + | − | − | − | − | + |
| Short stature | + | − | − | − | − | − | + |
| Microtia | + | − | + | − | − | − | + |

The expression of c-kit proto-oncogene is found in mast cells as well as in germ cell lineages. C-kit mutations potentially lead to a prolonged, chronic mastocytosis course by promoting mast cell growth and proliferation. Several c-kit variants have been described in association with the development of germ cell tumors, melanoma, neuroblastoma, and acute myeloid leukemia, as well as gastrointestinal stromal tumors [15–18,27–31].

The role of c-kit in the liver is controversial. Liver c-kit cells have been shown to participate in the liver repair processes, and c-kit mast cells have specifically been shown to play a role in the progression of liver fibrogenesis. C-kit is also a proto-oncogene and has been associated with primary liver cancers [43]. The role of Kit mutations in autoimmune hepatitis has not been described.

Our patient was unusual in several aspects. First, she presented with neonatal-onset SM, which is one of the rarest and most severe manifestations of congenital mastocytosis. Secondly, she developed a pure spinal germinoma that recurred within a very short period after the initial resection. She initially showed severe hepatomegaly with liver infiltration by mast cells, but then developed progressive liver fibrosis with cirrhosis and signs of portal hypertension. These clinical manifestations, as well as her hearing loss [44,45], could likely be attributed to a germinal KIT alteration. She also developed autoimmune hepatitis type 2, which as of now cannot be explained by the KIT variant.

Patients suffering from an aggressive SM, such as our patient, require therapy. As far as we know, there are limited data to guide the treatment for very young children presenting with this multisystemic disease. Imatinib was tried in early infancy for our patient. However, we now know that in the group of KIT-altered SM, Imatinib shows a low ORR of 0% [46] to 36% [47] and only 15% improved SM-related symptoms [48]. We do not know about the response of our patient to that treatment as she developed an acute, severe adverse event requiring the discontinuation of that drug. She then received Interferon, which had a favorable impact on the regression of the infiltration of mastocytosis into her liver. After a c-kit-related germ cell spinal neoplasm, Dasatinib was introduced.

Dasatinib is a multikinase inhibitor active against multiple genes such as *KIT*. However, while reports of Dasatinib therapy for systemic mastocytosis associated with the p.D816V variant suggest a reversal of disease progression in some patients, it does not appear that this treatment can eradicate all clinical manifestations [49,50]. Dasatinib has a great

in vitro activity response against various KIT mutations [50,51]. Results from a clinical study suggest that the KIT:c2447A > 7 pAsp816Val does not alter the binding of KIT by Dasatinib. This drug appears to inhibit Imatinib-resistant KIT activation loop mutants and induces apoptosis in human and murine mast cell lines carrying the KIT-D816V, 816F, and 816Y mutations.

Tyrosine kinase inhibitors, including Dasatinib and Imatinib, have shown potential in treating liver fibrosis in animal studies [52]. Sorafenib, used in the treatment of hepatocellular carcinomas in humans, has shown positive effects on cirrhosis in rats through its anti-fibrotic effect [53]. In humans, Sorafenib was shown to improve portal hypertension in cirrhotic patients. Several larger studies are on the way. Sorafenib has no role in mastocytosis as it induces mast cell maturation and skin reactions. Moreover, hepatotoxicity including acute liver failure has been described as a side effect of these agents, but autoimmune hepatitis has not been reported. In our patient, the long-term use of Dasatinib did not seem to prevent the progression of liver fibrosis. However, twelve years on from the start of Dasatinib treatment, mastocytosis involvement seems stabilized and no new c-kit neoplasm or germ cell tumor recurrence has been seen so far.

New emergent targeted therapies such as the use of Avapritinib, a selective KIT inhibitor with a high potency for the KIT D816V alteration, has been approved following the favorable results of two multi-center clinical trials for adults with advanced SM [54,55]. Other drugs such as Elenestinib (BLU-263), Masitinib, and Bezuclastinib are currently undergoing clinical trials for SM in adults. Additionally, a treatment algorithm guideline has been published for adults with SM [56].

## 7. Conclusions

We report the association of neonatal-onset SM, deafness, and spinal germinoma with the KIT pathogenic variant (NM_000222.3(KIT):c2447A > 7 pAsp816Val) in a patient who later developed multiple anomalies, including autoimmune hepatitis. This association could be linked to the conditions currently referred to as cutaneous mastocytosis, conductive hearing loss, and microtia (MIM 248910). Given that our patient harbors a KIT alteration, KIT is a plausible candidate gene for other patients presenting a similar phenotype, and if confirmed, this constellation would represent a KIT germline mutation syndrome.

This report extends the currently recognized spectrum of KIT-related disorders. It would be interesting to correlate the long-term outcome of this patient with that of others who also present with neoplasia and are reported to harbour KIT, and for whom tyrosine kinase inhibitor treatment will be undertaken.

**Author Contributions:** Formal investigations: V.L., A.W. and P.-O.G.; formal genetic analysis: N.J., I.D.B. and R.L.; pathology review: E.G.; manuscript writing: V.L., M.-F.P, P.-O.G., A.W. and I.D.B. All authors have read and agreed to the published version of the manuscript.

**Funding:** This research received no external funding.

**Institutional Review Board Statement:** Ethical review and approval were waived for this study because of the nature of this case report.

**Informed Consent Statement:** Written informed consent has been obtained from the parent of the patient to publish this paper.

**Data Availability Statement:** The data presented in this study are available on request from the corresponding author. The data are not publicly available due to privacy.

**Conflicts of Interest:** The authors declare no conflict of interest.

## Abbreviations

| | |
|---|---|
| ALT | Alanine Aminotransferase |
| BHCG | Beta-human chorionic gonadotropin |
| BM | Bone marrow |
| CGH | Comparative Genomic Hybridization |
| CSF | Cerebrospinal fluid |
| CT | Computed tomography scan |
| FISH | Fluorescence In Situ Hybridization |
| GGT | Gamma-Glutamyltransferase |
| MCs | Mast cells |
| MIM | Mendelian Inheritance in Man |
| MRI | Magnetic Resonance Imaging |
| SM | Systemic mastocytosis |
| WHO | World Health Organization |

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
