# Peer review of "A Review of the Clinical Features and Management of Systemic Congenital Mastocytosis through the Presentation of An Unusual Prenatal-Onset Case"

_curroncol, doi:10.3390/curroncol30100649_

Round 1

Reviewer 1 Report

This is a very interesting case, well worth publishing. 

Some comments:

1. WHO criteria for SM is included in the text. A reference to the last WHO criteria would have been sufficient. More interesting would be if you could include the criteria verifying the SM in the patient's  bone marrow biopsy (not only telling that this is SM)  since this is a very specific case, and we would really like to have the diagnosis verified by the criteria. 

I would also like to know the response of dasatinib since this nowadays is an uncommon treatment of patients with SM. Were the number of mast cells in the bone marrow reduced, were the clone size of KIT D816V reduced and was there a decrease in serum-tryptase? 

2. You are discussing treatment with imatinib and dasatinib. Imatinib is known to be effective in cases with wild type KIT mutation and a few other KIT mutations. It is not effective in cases with KIT D816V  (Pardanani A, AJH, 2021). Then no effect of imatinib could have been or can be excpected in your patient. On the other hand, I miss a discussion of alternative tyrosine kinase inhibitors. The newer and very potent selective tyrosine kinase inhibitors of KIT D816V, such as avapritinib, approved both in US and by EMA, and others in ongoing, studies such as BLU-263 and bezuclastinib, should be discussed.

Reviewer 2 Report

Larouche et al. describe a case of seemingly congenital c-kit (Asp816Val) mutation, leading to early onset systemic mastocytosis and germinoma later in life (two c-kit related malignancies). The case report is instructive. The concepts related to disease management, end-organ complications and associations with other conditions/syndromes are thoroughly covered. This work describing a rare condition will definitely be a relevant addition to the medical literature.  

In order to improve their work, the authors should consider the following:

1- The authors state that a skin biopsy confirmed the blood results establishing the de novo KIT mutation. Given the level of skin infiltration by mastocytes, how was ascertained the presence of the mutation in non-hematopoietic tissues at birth?

2-  Was does "...progression of real liver fibrosis was suspected" mean? Please clarify.

3- A figure showing the various findings in function of time (timeline) would help the reader (non-continuous chronological description of the findings as currently presented).

4- Pharmacological drug names should be used consistently (brand names used: topamax, imuran).

5- The description of syndromes overlapping with the case is interesting. Is there any genetic lesions associated with these entities? This should be specified. In addition, were other genetics lesions found in the patient? This should be presented/discussed.  

Only minors mistakes require correction. Careful revision should identify and correct grammar/syntax/spelling errors. 

The use of pronouns (she/her) instead of more general terms such as "patient" or "subject" may not be considered adequate, but this depends on editorial policies.  

Round 2

Reviewer 1 Report

Your case is very interesting. I have still several comments. You write that dasatinib has effect. I do not find any objective parametres that tell so. Response in the bone marrow? later development to mast cell leukemia does not indicate effect. Reduction of tryptase? Reduction of the KITD816V clone size? We know that the liver did not improve. Objective parametres are necessary to tell if there is response to dasatinib. Could there have been an extension of an IFN response? Then a stable disease? You seem to wonder about the development in the liver. With ASM in the liver fibrosis, portal hypertension and cirrhosis are common. Hepatitis developed. Dasatinib has hepatitis as a side effect. Have you considered that? I am wondering about if you have a dasatinib effect at all. If so, you have to prove that and tell us why. 

As mentioned, imatinib does not have effect in KITD816V pos.  SM. You have not included that in your discussion. You also write that most patients with SM require cytoreductive therapy.....and have KITD816V as a target, and the results have been disappointing in achieving CR. At first, patients with SM in generell do not require cytoreductive therapy. Patients with advanced SM may need that, but we will prefere to give patients with advanced SM tyroksine kinase inhibitors, not cytoreductive drugs. We cannot say that we obtain CR with the new tyrosine kinase inhibitors, but the number of mast cells are very much reduced. This part of the discussion should be changed.  

Read carefully. Some verbs and others words are missing

Reviewer 2 Report

The report is improved. No major concerns.

A final revision would be needed to correct remaining syntax errors (including in the revised text). 

Author Response

We have revised our manuscript accordingly. Thanks for your comments.